# Betulinic Acid Protects DOX-Triggered Cardiomyocyte Hypertrophy Response through the GATA-4/Calcineurin/NFAT Pathway

**DOI:** 10.3390/molecules26010053

**Published:** 2020-12-24

**Authors:** Jung Joo Yoon, Chan Ok Son, Hye Yoom Kim, Byung Hyuk Han, Yun Jung Lee, Ho Sub Lee, Dae Gill Kang

**Affiliations:** 1Hanbang Cardio-Renal Syndrome Research Center, Wonkwang University, 460, Iksan-daero, Iksan, Jeonbuk 54538, Korea; mora16@naver.com (J.J.Y.); hyeyoomc@naver.com (H.Y.K.); arum0924@naver.com (B.H.H.); shrons@wku.ac.kr (Y.J.L.); 2College of Oriental Medicine and Professional, Graduate School of Oriental Medicine, Wonkwang University, 460, Iksan-daero, Iksan, Jeonbuk 54538, Korea; 3Department of Ophthalmology, School of Medicine, Konkuk University, Gwangjin-gu, Seoul 05029, Korea; study0815@naver.com

**Keywords:** betulinic acid, doxorubicin, cardiomyocyte hypertrophy, apoptosis, heart failure

## Abstract

Cardiac hypertrophy is a major risk factor for heart failure and leads to cardiovascular morbidity and mortality. Doxorubicin (DOX) is regarded as one of the most potent anthracycline antibiotic agents; however, its clinical usage has some limitations because it has serious cardiotoxic side effects such as dilated cardiomyopathy and congestive heart failure. Betulinic acid (BA) is a pentacyclic-cyclic lupane-type triterpene that has been reported to have anti-bacterial, anti-inflammatory, anti-vascular neogenesis, and anti-fibrotic effects. However, there is no study about its direct effect on DOX induced cardiac hypertrophy and apoptosis. The present study aims to investigate the effect of BA on DOX-induced cardiomyocyte hypertrophy and apoptosis in vitro in H9c2 cells. The H9c2 cells were stimulated with DOX (1 µM) in the presence or absence of BA (0.1–1 μM) and incubated for 24 h. The results of the present study indicated that DOX induces the increase cell surface area and the upregulation of hypertrophy markers including atrial natriuretic peptide (ANP), B-type natriuretic peptide (BNP), beta-myosin heavy chain (β-MHC), and Myosin Light Chain-2 (MLC2) in H9c2 cells. However, the pathological hypertrophic responses were downregulated after BA treatment. Moreover, phosphorylation of JNK, ERK, and p38 in DOX treated H9c2 cells was blocked by BA. As a result of measuring the change in ROS generation using DCF-DA, BA significantly inhibited DOX-induced the production of intracellular reactive oxygen species (ROS) when BA was treated at a concentration of over 0.1 µM. DOX-induced activation of GATA-4 and calcineurin/NFAT-3 signaling pathway were remarkably improved by pre-treating of BA to H9c2 cells. In addition, BA treatment significantly reduced DOX-induced cell apoptosis and protein expression levels of Bax and cleaved caspase-3/-9, while the expression of Bcl-2 was increased by BA. Therefore, BA can be a potential treatment for cardiomyocyte hypertrophy and apoptosis that lead to sudden heart failure.

## 1. Introduction

Doxorubicin (DOX) is an anthracycline antineoplastic drug, which is commonly used for treatment to many kinds of cancers such as breast cancer, lung cancer, leukemias, lymphomas and several other carcinoma types [1]. However, administration of doxorubicin can be limited by side effects like immune suppression, vomiting, alopecia, and, most importantly, cardiotoxicity [2]. Cardiovascular complications are a major problem that restricts the widespread application of dose-dependent myocardial toxicity, resulting in irreversible congestive heart failure [3,4]. Cardiac hypertrophy is an independent risk factor for heart disease, such as hypertension, arrhythmias, myocardial infarction, and valvular insufficiency. In addition, it is also the main predisposing factor for heart failure and sudden cardiac death [5,6]. Accordingly, preventing the development of cardiac hypertrophy is conducive to reducing cardiovascular events. Cardiac hypertrophy is characterized by an increment in cardiomyocyte size including physiological and pathological hypertrophy [7]. Atrial natriuretic peptide (ANP), brain natriuretic peptide (BNP), β-myosin heavy chain (β-MHC), and α-skeletal actin are well known cardiac hypertrophy biomarkers [8,9].

ANP and BNP are secreted from cardiomyocytes by various factors and multiple signaling pathways regulate ANP and BNP. In heart failure, both plasma ANP and BNP increase due to compensatory homeostatic response to myocardial overload [10]. The circulating levels of ANP and BNP are positively related to ventricular dysfunction, and plasma levels of BNP are better reflecting markers of the severity of heart failure [11]. Mitogen-activated protein kinases (MAPKs) include three major subfamilies such as the extracellularly responsive kinases (ERKs), the c-Jun *N*-terminal kinases (JNKs), also known as stress-activated protein kinases (SAPKs), and the p38 MAPKs [12]. In the heart, intracellular MAPK signal pathways are well known to play a key role in the pathogenesis of cardiac hypertrophy in response to stimuli [13]. Reactive oxygen species (ROS) play a pivotal role in the development of pathological cardiac hypertrophy and heart failure. A recent study found that ROS production is the main trigger for the MAPK signaling pathway, a classical pathway involved in oxidative stress-induced hypertrophy [14,15].

GATA-4 is a member of the zinc finger transcription factor and plays an essential role in promoting myocardial differentiation and cardiac development as well as regulating the survival and hypertrophic growth of the adult heart [16]. GATA-4 has function as a key transcriptional regulator of numerous cardiac genes including ANP, BNP, α-myosin heavy chain (α-MHC), β-MHC, and many others [17]. During the hypertrophic stimulation, calcineurin dephosphorylated the nuclear factor of activated T-cells (NFAT) that may translocate into the nucleus to promote the gene expression, partly after forming a complex with GATA-4 [18]. In cardiac hypertrophy, NFAT is considered an important mediator of the signaling pathways involved in the coordinating pathological stimulation [19]. Activated NFAT has been reported to stimulate the expression of ANP and BNP and induces cell apoptosis [20]. Thus, the pathway of calcineurin/NFAT/GATA4 acts as an essential effector during cardiac hypertrophy formation.

Cardiomyocyte apoptosis is observed in the end stage human heart failure and is thought to play an important role in the development and progression of heart failure. Previous studies have shown that myocardial infarction caused by DOX plays a key role in the development of heart failure [21]. Additionally, more studies have shown that the programmed cell death, especially necroptosis and apoptosis, plays an important role in regulating the pathogenesis of DOX-induced cardiotoxicity by multiple signaling pathways, such as RIPK1/3 and Caspase-3, respectively [22,23]. The caspases are cysteinyl aspartate specific proteases that cleave target proteins at specific aspartate residues and play a central role in myocardial apoptosis and their activation occurs through cleavage at specific sites [24]. As an effector enzyme, caspase-3 is the major executors responsible for promoting cell apoptosis [25]. In addition, B-cell lymphoma 2 (Bcl-2) and Bcl2-associated X protein (BAX) are the major members of the Bcl-2 protein family that participate in mitochondrion-mediated apoptosis [26,27]. Bcl2 is an anti-apoptotic protein that regulates the function of the mitochondrial membrane to prevent cell death and blocks the activation of Caspase-3, while Bax is a pro-apoptotic protein and promotes cell apoptosis [28].

3β-Hydroxylup-20(29)-en-28-oic acid, also known as betulinic acid (BA), is a pentacyclic triterpene, and be derived chemically from betulin, a substance found in various plants, including *Quisqualis fructus*, *Coussarea paniculata*, *Caesalpinia paraguariensis*, *Vitex negundo*, *Berlinia grandiflora*, *Ziziphus joazeiro*, *Uapaca nitida*, *Ipomea pes-caprae*, *Ancistrocladus heyneanus*, *Diospyros leucomelas*, and *Syzygium claviforum* [29]. Many biological effects of BA have been described, including antitumor, anti-inflammatory, anti-angiogenic, and anti-fibrotic effects [30]. Recent studies have shown that BA protects against cerebral and renal ischemia reperfusion injuries [31]. However, the effect of BA on heart dysfunction by cardiac hypertrophy has not been demonstrated yet. Thus, in the present study, we investigated the protective effect of BA on DOX-induced cardiac hypertrophy and apoptosis in H9c2 cells.

## 2. Results

### 2.1. Effect of BA on DOX-Induced H9c2 Cell Death

For evaluation of BA pretreatment on DOX-induced cytotoxicity, H9c2 cells were pretreated for 30 min with BA (0.1–1 μM), then the medium was change and cells were treated with DOX (1 μM) for 24 h. The results of MTT assay and the iCELLigence microelectronic biosensor system system demonstrated that the viability of H9c2 cells decreased significantly after incubation at 1 µM DOX for 48 h (*p* < 0.01). Pretreatment with BA increases cell viability reduced by DOX in a dose-dependent manner (Figure 1A,B). We also confirmed the effect of BA on cell death through real-time cell image measurement using the Lionheart FX Automated Microscope. The observation of H9c2 live cell imaging showed that cytotoxicity of DOX increased by BA treatment in a dose-dependent manner (Figure 1C). Thus, BA treatment mitigates the cytotoxicity of H9c2 under DOX conditions.

### 2.2. Effect of BA on DOX-Induced Cardiac Hypertrophy

H9c2 cell exposed to 1 μM DOX for 24 h exhibited a significant increase in cell surface area by 2.64 folds compared with controls (*p* < 0.01). However, pretreatment of BA ≥ 0.5 μM blocked the DOX–induced increase in cell size (2.64 ± 0.1891 versus 1.9060 ± 0.2172 (*p* < 0.05), 1.3600 ± 0.1242 (*p* < 0.01), Figure 2A). ANP, BNP, β-MHC, and MLC-2v are markers of cardiac hypertrophy and heart failure. To determine whether BA suppresses the markers for cardiac hypertrophy, we performed Western blotting and real-time reverse transcription-polymerase chain reaction (RT-PCR). As shown in Figure 2B,C, protein expressions of ANP, BNP, β-MHC, and MLC-2v were significantly increased by DOX. However, treatment with BA significantly decreased the expression of hypertrophy biomarkers protein (Figure 2B). Similarly, BA significantly inhibited DOX-induced ANP, BNP, β-MHC, and MLC-2v mRNA levels in H9c2 cells (Figure 2C). Thus, these results suggest that BA treatment regulates expression of cardiac hypertrophy marker genes.

### 2.3. Effect of BA on DOX-Induced MAPK/ROS Signaling Pathways

MAPK is one of the signaling pathways involved in cardiac hypertrophy and heart failure [32]. We examined the effects of BA on the mitogen-activated protein kinase (MAPK) signaling pathway. As shown in Figure 3, DOX enhanced the phosphorylated expression level of JNK, ERK, and p38 MAPK and that these changes were blocked by high-dosage BA treatment (*p* < 0.01). ROS production is known to play a major role in triggering the MAPK signaling pathway. Oxidative stress has been reported to be essential for cardiac hypertrophy development [33]. As shown in Figure 4, ROS generation were markedly up-regulated by DOX, while being down-regulated by BA ≥ 0.5 μM or ROS scavenger (*N*-acetyl-cysteine, NAC) pre-treatment. Thus, these results suggesting that BA inhibits cardiac hypertrophy by blocking ROS-dependent MAPK signaling.

### 2.4. Effect of BA on GATA4 Activation

Following DOX administration, the activation of transcription factor GATA-4 involved in expression of cardiac hypertrophy was investigated. The results showed that phosphorylated level of GATA-4 was highly expressed by DOX. However, BA effectively prevented DOX-induced p-GATA-4 protein expression (Figure 5A). In addition, immunofluorescence assay was performed to determine the effect of BA on DOX-induced the nuclear translocation of GATA-4 phosphorylation. As shown in Figure 5C, pretreatment of BA inhibited the activation of GATA-4 compared with DOX only treatment. Therefore, the results indicate that BA treatment inhibited DOX-induced cardiac hypertrophy through blocking of GATA-4 activation, the transcriptional regulator for the generation of cardiac hypertrophy.

### 2.5. Effect of BA on Calcineurin Related Pathway

Calcineurin is known as a hypertrophy signaling factor based on overexpression in the heart. The calcineurin–NFAT signaling is activated from pathological cardiac hypertrophy and heart failure. In the present study, BA reduced DOX-induced calcineurin protein expression in H9c2 cells (Figure 6A). In addition, we also examined the protein expression levels of NFAT-3 in cytoplasm and nucleus following DOX treatment (Figure 6B). As shown in Figure 6D, pretreatment of BA reduced the nuclear level of NFAT-3 in a dose-dependent manner and was accompanied by the relative induction of cytosolic level of NFAT-3. In this study, effect of BA on nuclear translocation of NFAT-3 was further confirmed by immunofluorescence assay in H9c2 cells exposed to DOX. The nuclear localization of NFAT-3 improved significantly when cells were treated with BA (Figure 6C). Therefore, these results suggest that BA improves cardiac hypertrophy by regulating calcineurin-NFAT-3 signal.

### 2.6. Effect of BA on H9c2 Cell Apoptosis

Apoptosis has been implicated in both acute and chronic heart diseases. To clarify the effect of BA on apoptosis in H9c2 cells exposed to DOX, we performed Western blotting and real-time reverse transcription-polymerase chain reaction (RT-PCR). The expression levels of cardiac apoptosis-related proteins showed that DOX increased the expression of pro-apoptotic factors, Bax and cleaved caspase-3/-9, whereas Bcl-2, an anti-apoptotic indicator, decreased after DOX treatment. However, pre-treatment with BA decreased Bax and cleaved caspase-3/-9 expression and induced Bcl-2 expression in H9c2 cells (Figure 7A,B). Meanwhile, DOX increased Bax and AhR mRNA expression, while BA inhibited that (Figure 7C). In the present study, effect of BA on DOX-induced cardiac apoptosis was further confirmed by Annexin V-FITC/PI dual staining and flow cytometry were used. As shown in Figure 7D, BA treatment improved the apoptosis rate compared to DOX alone treatment (*p* < 0.01). Thus, these results suggest that BA treatment improves DOX-induced cardiac apoptosis through the regulation of apoptosis-related factors, which further improves heart failure caused by cardiac hypertrophy.

## 3. Discussion

Cardiac hypertrophy is a major risk factor for cardiovascular diseases, such as hypertension, arrhythmia, and myocardial infarction. In particular the risk of heart failure increases significantly due to prolonged cardiac hypertrophy, which increases the risk of cardiovascular mortality [5,6]. Cardiac hypertrophy is characterized by increased cell size and enhanced protein synthesis, embryonic gene ANP, BNP and β-MHC expression [7].

BA, a pentacyclic lupane type triterpene, can be extracted from various plants, including birch trees (*Betula alba*). Many biological effects of BA have been reported to have antitumor, anti-inflammatory, anti-angiogenic, and anti-fibrotic effects. Recent studies have shown that BA protects against cerebral and renal ischemia reperfusion injuries [30,31]. DOX is an anthracycline chemotherapy drug, commonly used in various cancer treatments; however, the use of DOX is limited by side effects like cardiotoxicity, comprising cardiomyopathy and ultimately fatal congestive heart failure [1,2]. However, very little is known about the role of BA in cardiac hypertrophy induced by DOX. In the present study, we investigated the protective effect of BA against DOX-induced cardiac hypertrophy and apoptosis in H9c2 cells.

In this study, we first explored the effect of BA on the cardiotoxicity induced by DOX in H9c2 cells determined by MTT assay and real-time cell live imaging analysis. DOX cardiotoxicity features apoptosis or other forms of cell death in cardiomyocytes, resulting in loss of functional myocytes and irreversible heart injury. In general, cardiac hypertrophy occurs thorough the enlargement of cell size. Consistent with previous reports, our results demonstrated that exposure of H9c2 cells to DOX leads to decrease Cell viability and Cell Index. This result was similarly found in the results of measuring real-time cell live images. However, BA pretreatment up-regulated the reduced-cell viability and cell index. We further measured the change of cell surface area by F-actin staining. The results illustrated that a cell surface area enlargement was observed in H9c2 cells treated with DOX for 24 h, while BA down-regulated the cell surface area enlargement in DOX-induced H9c2 cells. This data supports that BA can alleviate cardiac hypertrophy through improved DOX-induced cardiotoxicity.

Common downstream readouts of induction of pathological cardiac hypertrophy, both in vivo and in vitro, include reactivation of a fetal gene program. Activation of fetal genes such as β-MHC, ANP, BNP and α-skeletal muscle actin is a hallmark of cardiac hypertrophy [34,35]. Our results showed that BA treatment strongly ameliorated the expression of hypertrophy biomarkers protein induced by DOX administration as demonstrated by Western blot assay. Similarly, BA pretreatment lowered the elevated ANP, BNP, β-MHC, and MLC-2v mRNA expression in DOX-induced H9c2 Cells.

ROS can activate MAPK signaling pathways such as p38, ERK, and JNK, which plays a pivotal role in cardiovascular diseases. The activation of MAPK signaling pathways is closely related to ROS generation and the cardiac hypertrophy [36]. In this study, DOX significantly enhanced the phosphorylated expression levels of p38, ERK, and JNK MAPK. BA protects H9c2 cells by suppressing the DOX-induced activities of p38, ERK and JNK MAPK phosphorylation expression. Furthermore, we found that ROS generation was markedly up-regulated by DOX, while being down-regulated by BA ≥ 0.5 μM or ROS scavenger (NAC) pre-treatment. These results suggest that BA can have a myocardial protection effect on DOX-induced cardiac hypertrophy through inhibition of the ROS/MAPK pathway.

Previous studies have clearly demonstrated that pharmacological inhibition of this hypertrophy signaling pathway is important in curving hypertrophy and heart failure [37]. Calcineurin is involved in the signaling pathways leading to myocardial apoptosis and heart hypertrophy. It has also been demonstrated that calcineurin/NFAT signaling pathway plays an important role in the development of cardiac hypertrophy and end-stage heart failure [38]. In the nucleus, NFAT-3 activated in the nucleus interacts closely with GATA-4 [39], subsequently turning on myocardial genes during hypertrophy. The transcription factor GATA-4 acts as a downstream part of the activated calcineurin signaling pathway, plays an important role in cardiac adaptive responses including cardiac hypertrophy and survival [40]. The results of recent study indicated that DOX deplete the transcription factor GATA4 in cardiomyocytes [41], and restoration of GATA-4 levels prevents DOX-induced myocardial hypertrophy and cell death [42]. However, further study needs to be given to whether DOX is sufficient to activate calcineurin and subsequently causes myocardial disease. Western blot assay and immunofluorescence staining also demonstrated that DOX induces cardiac hypertrophy through GATA-4 and calcineurin/NFAT-3 signaling pathway, while pretreatment of BA reduced the p-GATA-4 and NFAT translocation as well as inhibited the calcineurin expression in DOX-induced H9c2 cells. Therefore, these results suggest GATA-4 and calcineurin/NFAT-3 signaling pathway are required to mediate the development of cardiac hypertrophy by DOX. It also shows that BA can reduce the DOX-induced hypertrophy effect by blocking the GATA-4 and calcineurin/NFAT-3 signaling pathways.

Reportedly, for cardiomyocyte hypertrophy and heart failure, the intrinsic caspase-mediated pathway is important [43]. The activation of caspase-3 with the change of mitochondrial transmembrane potential is considered one of the pivotal stages during the apoptosis [44]. Bcl-2 as an anti-apoptotic protein plays an important role in maintaining the mitochondrial structure and function and inhibiting cell apoptosis is a potent downward regulation of Bcl-2 causing heart failure. However, Bax encode pro-apoptotic proteins that can promote cytochrome c release from the mitochondria to induce apoptosis [45]. Accumulating evidence shows that cardiomyocyte apoptosis and subsequent cell loss caused by DOX treatment resulted in cardiac function disorder proven by the elevated expression of Bax, cleaved Caspase-3, as well as the reduced Bcl-2 levels [46]. Western blot analysis of the expression levels of apoptosis showed that DOX increased the expression of pro-apoptotic factors, Bax and cleaved caspase-3/-9 and caspase-8 while Bcl-2, an anti-apoptotic indicator, decreased after DOX treatment. Treatment with BA could markedly attenuate apoptosis in H9C2 cells stimulated by DOX. Additionally, to determine whether BA affects cell viability, we examined two types of cell apoptosis by flow cytometry analysis with Annexin V-PI staining under DOX conditions. PI staining analysis showed that DOX showed significantly increased H9c2 apoptosis and treatment with BA markedly reduced H9c2 apoptotic cell death. Therefore, these results indicated that the inhibitory effect of BA in H9c2 cells was associated with the induction of apoptotic cell death through regulation of several major growth regulatory gene products such as Bcl-2/Bax family expression and caspase protease activity. However, our current research has several limitations. The H9c2 cell line used in this study, although possessing some characteristics of cardiomyocytes, do not fully conform to primary cardiomyocytes, and further exploration should be done using primary neonatal cardiomyocytes or in vivo experiments. In addition, more specific markers for hypertrophic hearts should be considered in the future as it has been recently studied that protein expression of ANP and BNP does not adequately indicate hypertrophy in vivo. Therefore, additional research is needed to clarify the molecular and cellular mechanisms involved in anti-hypertrophic activity of BA.

## 4. Materials and Methods

### 4.1. Chemicals

Betulinic acid was purchased from Calbiochem (San Diego, CA, USA). Dulbecco’s Modified Eagle’s Medium (DMEM), Fetal Bovine Serum (FBS), and penicillin-streptomycin F-actin, Alexa Four 488 phalloidin, dichlorofluorescin diacetate (DCFH-DA), DAPI were purchased from Thermo Fisher Scientific (Waltham, MA, USA). Doxorubicin, 3-(4,5-Dimethylthiazol-2-yl)-2,5-diphenyl tetrazolium (MTT), primary antibodies for p38, ERK, JNK, p-ERK1/2, p-p38, NfκB p65, Bcl-2, Bax, β-actin (sc-47778), β-MHC, MLC-2v, p-GATA-4, GATA-4, Calcineurin, HDAC, LaminB, and HRP conjugated secondary antibodies raised against mouse, rabbit and goat were purchased from Santa Cruz Biotechnology (Santa Cruz, CA, USA). NFAT-c3, Caspase-3, Caspase-9, Bax, and α-tubulin were purchased from Cell signaling technology (Danvers, MA, USA). ANP and BNP were purchased from ABCam (Danvers, MA, USA).

### 4.2. Cell Culture

The rat H9c2 cells were procured from American Type Culture Collection (Manassas, VI, USA) and were cultured in DMEM containing 10% FBS and antibiotics (1% penicillin streptomycin) at 37 °C with humidified atmosphere of 95% air and 5% CO_2_. Cells were sub-cultured at 80% confluency and media was changed once in two days. All the experiments were performed within 20–40 passages in order to confirm cell population, uniformity and reproducibility. The BA was treated 30 min prior to DOX exposure and subsequent stimulation with DOX for 24 h. The cells were then harvested and used in the following experiments.

### 4.3. Cell Viability and Cell Index Assay

The cell viability was determined using a modified 3-(4,5-dimethylthiazol-2-yl)-2,5-diphenyl tetrazolium (MTT) assay. Briefly, the H9c2 cardiomyocytes were plated in 96-well plates with cell population of 1 × 104 cells/well and allowed for attachment overnight at 37 °C in CO_2_ incubator. H9c2 cells were pre-treated with BA (0.1, 0.5 and 1 μM) for 30 min and DOX (1 μM) for 24 h. Then, 10 μL of MTT solution (1 mg/mL) was added to each well and incubated for 4 h in CO_2_ incubator. After 4 h, the formazan crystals precipitate was dissolved by adding 100 μL of DMSO. The absorbance at 540 nm was measured using a spectrofluorometer (F-2500, Hitachi, Tokyo, Japan). Mesangial cell index was calculated for each E-plate well by RTCA Software 1.2 (Roche Diagnosis, Boulogne-Billancourt, France). The graphs were real-time generated outputs from the iCELLigence system. The morphological changes and real-time cell live imaging of cells were observed using the Lionheart FX (BioTek, Winooski, VT, USA). The culture plates were placed in the Lionheart FX Live-cell imaging microscope, equipped with full temperature and CO_2_ control to maintain 37 °C and 5% CO_2_. Z-stack images were acquired 1, 5, 10, and 21, 23, and 24 h after plating for 5 fields of view per well at 20× magnification. Automated image capture and movie production were performed using Gen5 ™ 3.0 software ((BioTek, Winooski, VT, USA).

### 4.4. Cell Surface Area Measurement

The cell surface area, as a hallmark of hypertrophy, was quantified using F-actin staining. In brief, H9c2 cells were then fixed with 4% paraformaldehyde and then 0.1% Triton X-100 was added to the cells at room temperature for 5 min. Afterwards, cells were blocked with 1% BSA for 1 h. After washing with PBS three times, the cells were stained using phalloidin-Alexa488 for 30 min to visualize F-actin, and then counterstained with DAPI. The cells were imaged with a fluorescence microscope and the cell size was analyzed using EVOS-M5000 Cell Imaging System (Thermos fisher scientific, Waltham, MA, USA).

### 4.5. Western Blot Analysis

After treatment with the indicated condition, cells were harvested and washed with cold phosphate-buffered saline (PBS) and followed by incubation in radioimmunoprecipitation (RIPA) buffer in 4 °C for at least 30 min. The cell lysates were clarified by centrifugation at 12,000 rpm for 10 min at 4 °C and the supernatants were collected. Cells lysates (30 μg) were separated on 10% SDS-polyacrylamide gel electrophoresis and transferred to nitrocellulose paper. Blots were then washed with H2O, blocked with 5% skimmed milk powder in TBST [10 mM Tris-HCl (pH 7.6), 150 mM NaCl, 0.05% Tween-20] for 1 h and incubated with the appropriate primary antibody at dilutions recommended by the supplier. Then the membrane was washed, and primary antibodies were detected with goat anti-rabbit-IgG conjugated to horseradish peroxidase, and the bands were visualized with enhanced chemiluminescence (Amersham, Buckinghamshire, UK). Protein expression levels were determined by analyzing the signals captured on the nitrocellulose membranes using the iBright™ FL1000 (Thermos Fisher Scientific, Waltham, MA, USA).

### 4.6. Real-Time PCR

Cell were collected and RNA extracted using the TRIzol^®^ reagent (Invitrogen, Carlsbad, CA, USA). About 1 μg of RNA was reverse transcribed into cDNA using for the Reverse Transcription Master Premix (ELPISBIO, Korea). The PCR amplification was conducted with a TOPreal™ qPCR 2X PreMIX (SYBR Green with high ROX, enzynomics) using Applied Biosystems real-time PCR system (Applied Biosystems, Foster city, CA, USA). Specific sense and antisense primers used were as follows respectively: ANP, sense: 5′-GAG GAG AAG ATG CCG GTA G-3′, anti-sense: 5′-CTA GAG AGG GAG CTA AGT G-3′; BNP, sense: 5′-TGA TTC TGC TCC TGC TTT TC-3′, anti-sense: 5′-GTG GAT TGT TCT GGA GAC TG-3′; β-MHC, sense: 5′-GCA TTC TCC TGC TGT TTC CT-3′, anti-sense: 5′-CCC AAA TGC AGC CAT CTC-3′; MLC-2v, sense: 5′-CCT AAC GTC ACC GGC AAC C-3′, anti-sense: 5′-TTT GGT TCA CAT CAT CAC CCA-3′; GATA-4, sense: 5′-GGG CGA GCC TGT TTG CAA TG-3′, anti-sense: 5′-TGC TTG GAG CTG GCC TGT GA-3′; Bax, sense: 5′-AGA CAC CTG AGC TGA CCT TGG A-3′, anti-sense: 5′-CGC TCA GCC GCT CAG CTT CTT GGT GGA T-3′; AhR, sense: 5′-CGG CAG ATG CCT TGG TCT TCT ATG C-3′, anti-sense: 5′-TGG AAC TCA GCT CGG TCT TCT GTA TGG-3′; α-tubulin, sense: 5′-GAC CAA GCG TAC CAT CCA GT-3′, anti-sense: 5′-CCA CGT ACC AGT GCA CAA AG-3′. The data are presented as the fold change in gene expression normalized to the endogenous reference gene (α-tubulin) and relative to the untreated control.

### 4.7. Measurement of ROS

The fluorescent probe CM-H_2_DCFDA was used to measure the intracellular generation of ROS. Briefly, the confluent cell in the 96 well culture plates were pretreated with BA for 30 min and stimulated in absence or presence of DOX for 24 h. The cells were incubated at 37 °C with 10 µM CM-H_2_DCFDA. Fluorescence intensity was measured by fluorescence microscopy (Infinite F200 pro (TECAN, Männedorf, Switzerland): excitation 488 nm, emission 513 nm) and examined under a fluorescence microscope (Eclipse Ti, Nikon, Tokyo, Japan).

### 4.8. Immunofluorescence Microscopy

H9c2 cells on glass coverslips were fixed in 4% paraformaldehyde for 30 min and then permeabilized with 0.1% Triton X-100 for 5 min in PBS, washed 3 times with PBS. Non-specific binding of the fixed cells was blocked with PBS containing 1% BSA at 37 °C for 30 min. Afterwards, samples were incubated with primary antibody (GATA-4 and NFAT-3) at 4 °C overnight. Corresponding secondary antibodies were labeled with Alexa Fluor 488 (1:200; Molecular Probes, Eugene, OR, USA). DAPI (4′,6-diamidino-2-phenylindole) was used for nuclear staining. The cells were then observed under a fluorescence microscope, EVOS-M5000 Cell Imaging System (Thermos fisher scientific, Waltham, MA, USA).

### 4.9. Flow Cytometry of Apoptosis

Apoptosis was measured by using Annexin V-FITC/propidium iodide (PI) double-staining apoptosis detection kit (Cayman chemicals, Ann Arbor, MI, USA) and flow cytometry. In brief, The H9c2 cells were collected by trypsinization and centrifuged at 1200 rpm for 5 min. Following suspension in binding buffer, cells were labelled with Annexin-V-FITC and Propidium Iodide (PI) for 15 min at 37 °C in the dark. Finally, cell samples were quantified by Attune™ NxT Flow Cytometer (Thermos fisher scientific, Waltham, MA, USA). The index of apoptosis was expressed as the percentage of total apoptotic cells, which included the percentage of early apoptotic cells (Annexin V positive and PI negative) plus the percentage of late apoptotic cells (Annexin V positive and PI positive).

### 4.10. Statistical Analysis

All the experiments were repeated at least three times. The results were expressed as a mean ± S.E., and the data were analyzed using one-way ANOVA followed by a Dunnett’s test or Student’s *t*-test to determine any significant differences. *p* < 0.05 was considered as statistically significance.

## 5. Conclusions

In conclusion, the results of the present study demonstrated that BA protects H9c2 cells from DOX-induced hypertrophy and apoptosis at least partly through the inhibition of ROS/MAPK and GATA-4/calcineurin/NFAT signaling pathway. Thus, these results show that BA has cardio-protective effects and has the potential as a new therapeutic agent for the treatment of cardiac hypertrophy and ventricular dysfunction leading to heart failure.

## Figures and Tables

**Figure 1 molecules-26-00053-f001:**
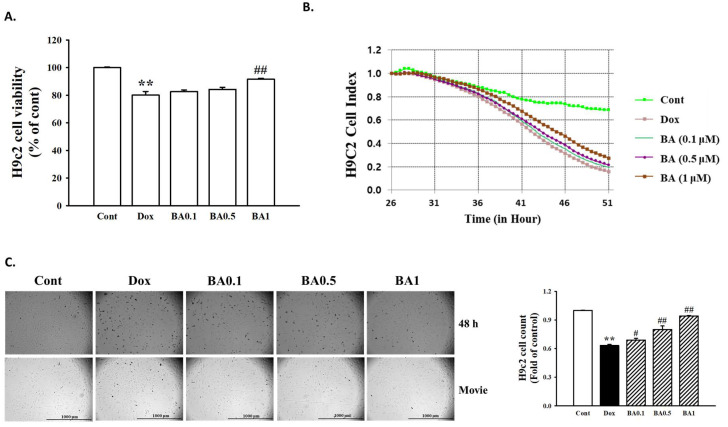
Effect of betulinic acid (BA) on doxorubicin (DOX)-induced H9c2 cell death. (**A**) Effects of BA on DOX-induced changes of cell viability. H9c2 cells were pretreatment with different concentrations of BA for 30 min prior to DOX stimulation, and the cell viability was detected by 3-(4,5-dimethylthiazol-2-yl)-2,5-diphenyl tetrazolium (MTT) assay. (**B**) The results of H9c2 cell Index using xCELLigence RTCA DP Real time cell analyzer. (**C**) Observation of H9c2 live cell imaging under the Lionheart FX Automated Microscope (Scale bar = 1000 μm). Representative images and quantitative results demonstrating that BA (0.1–1 μM) inhibited DOX (1 μM; 48 h)-induced H9c2 cells. All experiments were performed at least three times. Data represent mean ± SD. ** *p* < 0.01, vs. control; # *p* < 0.05, ## *p* < 0.01, vs. DOX-treated cells.

**Figure 2 molecules-26-00053-f002:**
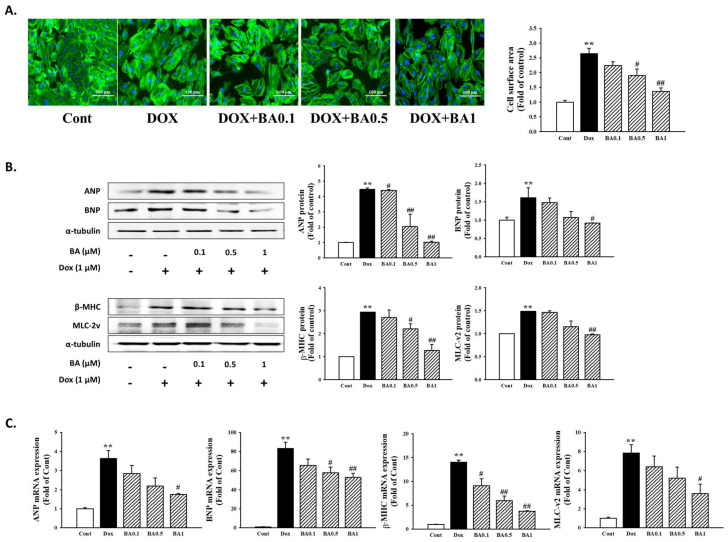
The effects of BA on DOX-induced cardiac hypertrophy. (**A**) The effects of BA on cell surface area size. The cell surface area was measured using anti-F-actin staining (green) under fluorescence microscopy. The nucleus was stained with DAPI (blue) (Scale bar = 100 μm). (**B**,**C**) Effect of BA on cardiomyocyte hypertrophy markers in DOX-treated H9c2 cells. The relative expression of ANP, BNP, β-MHC, and MLC-2v was determined using Western blot analysis and RT-qPCR assay. All experiments were performed at least three times. Data represent mean ± SD. ** *p* < 0.01, vs. control; # *p* < 0.05, ## *p* < 0.01 vs. DOX-treated cells.

**Figure 3 molecules-26-00053-f003:**
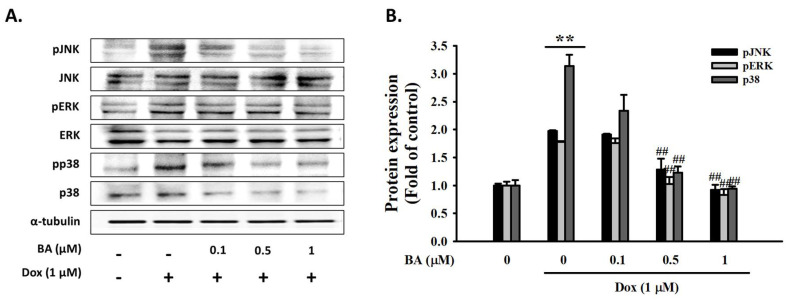
The effects of BA on the expression of MAPK markers. Protein expression of MAPK markers (JNK, ERK, and p38) analyzed by Western blot. (**A**) Representative blots and (**B**) quantitative results demonstrating that BA decreased the phosphorylated levels of JNK, ERK, and p38 in response to DOX. The results are expressed as the mean ± SE values of three experiments. ** *p* < 0.01, vs. control; ## *p* < 0.01, vs. DOX-treated cells.

**Figure 4 molecules-26-00053-f004:**
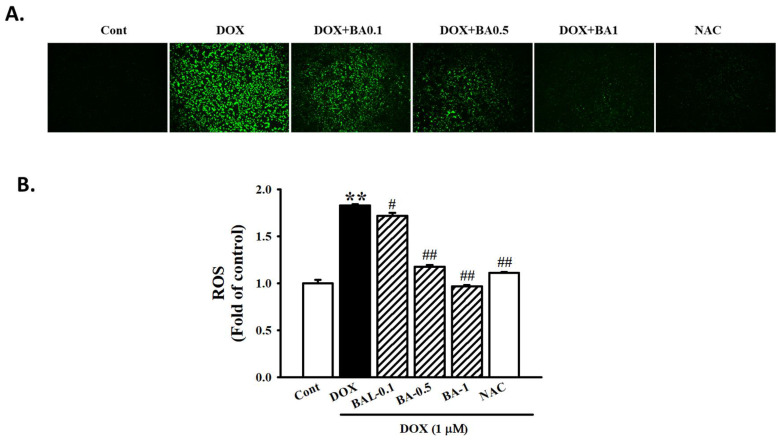
Effect of BA on ROS formation in DOX-induced cardiac hypertrophy. (**A**) Analysis of ROS formation using DCFH-DA staining on H9c2 cells treated DOX and pre-administration of BA after 30 min of treatment. (**B**) Alternatively, the relative ROS levels were quantified by fluorescence microplate. The images were taken using light (magnification 20×). All experiments were performed at least three times. Data represent mean ± SD. ** *p* < 0.01, vs. control; # *p* < 0.05, ## *p* < 0.01, vs. DOX-treated cells.

**Figure 5 molecules-26-00053-f005:**
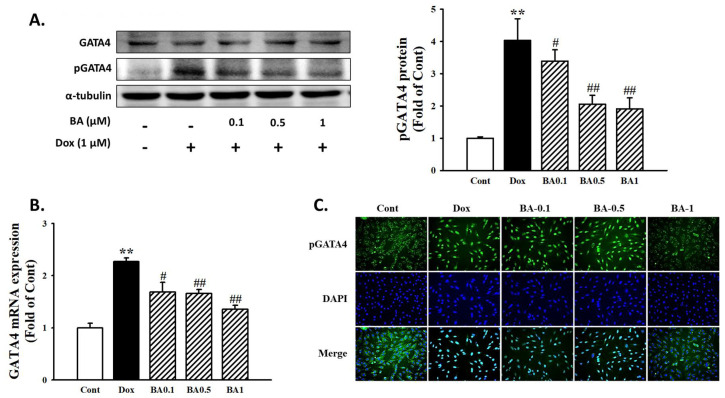
The effects of BA on activation of GATA4. (**A**) The protein levels of GATA-4 and phosphorylated GATA-4 were determined by Western blot analysis. (**B**) GATA-4 mRNA level was analyzed by using Real-time PCR. (**C**) Immunofluorescent images of p-GATA-4 nuclear translocation under the laser scanning confocal microscopy were show (magnification 400×). Nuclei were stained with DAPI (blue) and p-GATA-4 was stained with Alexa Fluor 488 (green) (Immunofluorescence, 200×). The results are expressed as the mean ± SE values of three experiments. ** *p* < 0.01, vs. control; # *p* < 0.05, ## *p* < 0.01, vs. DOX-treated cells.

**Figure 6 molecules-26-00053-f006:**
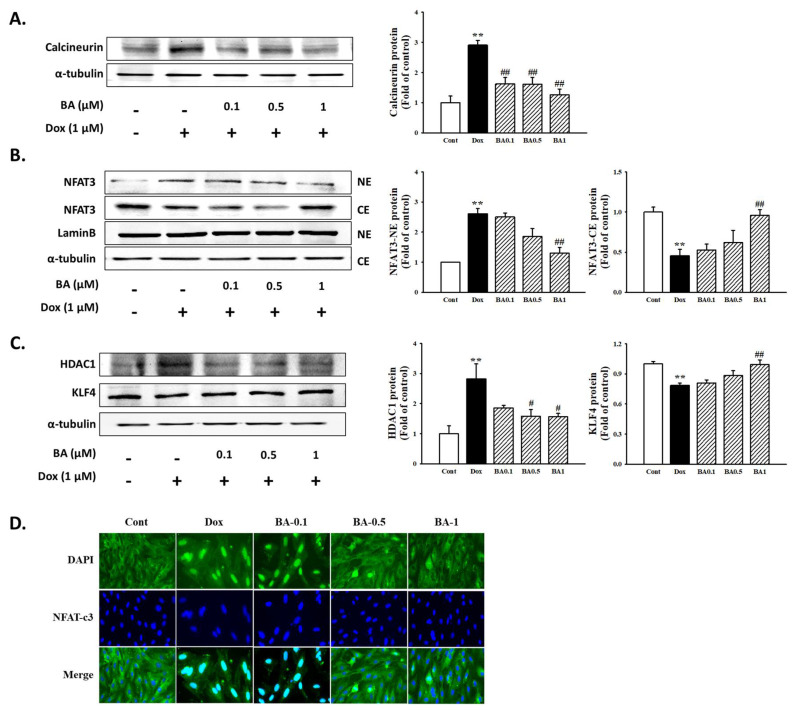
The effects of BA on calcineurin/NFAT-3 signaling pathway. Expression of calcineurin (**A**), nuclear localization of NFAT-3 (**B**), and HDAC1/KLF4 (**C**), were determined by Western blot analysis. (**D**) Nuclear translocation of NFAT was examined by immunofluorescence analysis (Immunofluorescence, 400×). The results are expressed as the mean ± SE values of three experiments. ** *p* < 0.01, vs. control; # *p* < 0.05, ## *p* < 0.01, vs. DOX-treated cells.

**Figure 7 molecules-26-00053-f007:**
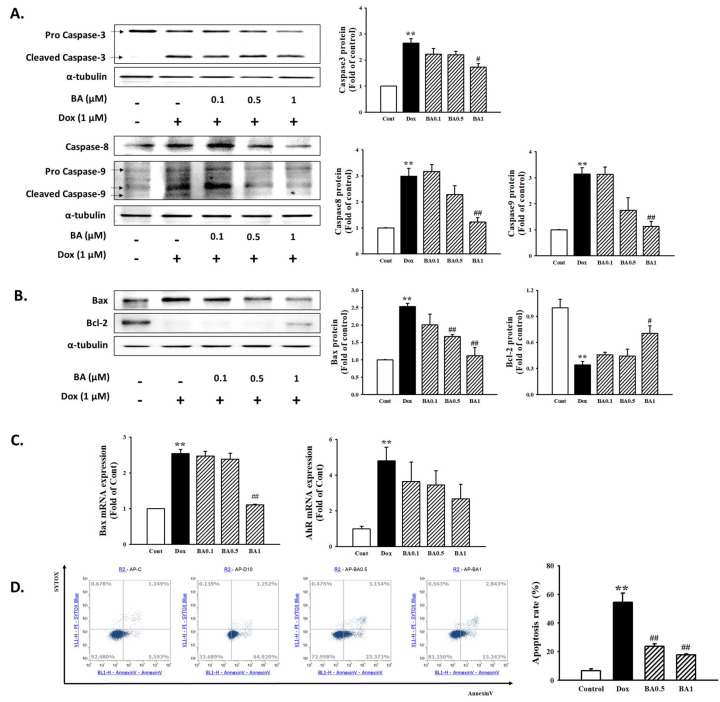
Effect of BA on cell apoptosis in DOX-induced cardiac hypertrophy. (**A**) Western blot analysis results showing the expression levels of apoptosis-related proteins including Caspase-3 and Caspase-9 in DOX-induced H9c2 cells. (**B**) The protein and mRNA expression of Bax and Bcl-2 was examined by Western blot analysis. (**C**) Bax and AhR mRNA expression was examined by real time PCR (**D**) Cell apoptosis was measured using Annexin-V/PI staining and flow cytometry. All experiments were performed at least three times. Data represent mean ± SD. ** *p* < 0.01, vs. control; # *p* < 0.05, ## *p* < 0.01, vs. DOX-treated cells.

## Data Availability

The datasets used and/or analyzed during the current study are available from the corresponding author upon reasonable request.

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
