# Peer review of "Betulinic Acid Protects DOX-Triggered Cardiomyocyte Hypertrophy Response through the GATA-4/Calcineurin/NFAT Pathway"

_molecules, 2020, doi:10.3390/molecules26010053_

Round 1
Reviewer 1 Report
Manuscript by Yoon et al looks at the ability of betulinic acid to prevent injury to a cardiomyocyte cell line. Manuscript was well written.
- Title should be rewritten as the study is performed in a cardiac cell line and not in vivo. Can talk about cardiomyocyte cell line but not cardiac injury. H9c2 are a good start but are far from mature cardiomyocytes and this limitation should be acknowledged.
- In a similar way, cardiac hypertrophy is not accurate for this study as isolated cells are used. Cardiomyocyte hypertrophy can be used.
- There appears to be a disconnect between Figures 1A and 1B. In 1A, BA rescued significantly cell viability but this degree o rescue is not evident in Figure 1B. In fact I don’t think there is a significant difference between BA 1 um and Dox at 51 hours. Error bars should be shown for 1B. For 1A, the Y axis should not start at 70 percent but start at 0 so the magnitude of the changes observed is more accurately depicted.
- Western blot for figure 2B is oversaturated and has too high a contrast. Lanes don’t line up either. For bottom part of 1B are the lanes from separate blots? Appears to be. All blots should be from the same blot.
- Higher power image of H92C cells should be shown for Figure 2A
- Figure 3. Representative blots should be from the same blot and not separate lanes. Quality of blot is not very good.
- For Figure 4, the y axis should be continuous to reflect the actual differences more clearly for the reader.
- Figure 5C the panels are mislabeled.
- Not cardiac apoptosis in H29C but apoptosis in H29C
- Western blots of Figure 7 are not clear and appear to spliced together for the bottom panels. As mentioned before, should be from a single blot or clearly marked as from separate blots.
Reviewer 2 Report
Betulinic Acid Protects DOX-indued Cardiac injury by down-regulation of Hypertrophy and Apoptosis.
Jung Joo Yoon et al.
The purpose of this work was investigate the effect of betulinic acid (BA) on doxorubicin (DOX)-induced cardiac hypertrophy and apoptosis in H9c2 (rat myoblastic) cells through analyses of cell viability, cell area surface, expression (western blot and mRNA) of hypertrophy markers, MAPK signaling (p38, ERK, and JNK expression), reactive oxygen species, transcription factor GATA4, calcineurin pathway and apoptosis process.
Data obtained support the following conclusion: BA protects H9c2 cells from DOX-induced hypertrophy and apoptosis at least partly through the inhibition of ROS/MAPK and GATA-4/calcineurin/NFAT signaling pathway.
According to these results BA may be considered as a new potential agent for alleviate cardiac hypertrophy and ventricular dysfunction that can lead to heart failure.
OBSERVATIONS.
Why the authors study calcineurin/NFAT pathway? Cardiac hypertrophy is regulated by different signaling pathways as calcineurin/NFAT pathway and insulin-like growth factor-I–phosphatidylinositol 3-kinase–Akt/protein kinase and in this process participates also transcription factors such as NFAT, myocyte enhancer factor-2 (MEF-2), GATA4/6 and NFκB.
Quantifications of the following figures is needed: figure 1C (cell live images), figure 2A (cell surface images), figures 2B and 2C (western blot images), figure 5A (western blot of GATA4 forms); Figures 6A, B and C (western blot of calcineurin, NFAT3 and HDAC1/ KLF4, respectively), and figures 7A and B (western blot of apoptosis markers).
Western blots images for ANP y BNP show that BA 1 uM/Dox 1 uM treatments reduce their expression levels even below of control cells. Explain these facts and their possible consequences.
Section 2.3. Respect to ROS and MAPK signaling pathway the authors indicate the following: “Thus, these results suggesting that BA inhibits cardiac hypertrophy by blocking ROS-dependent MAPK signaling.” Why MAPK signaling changes were associated with changes in ROS levels? It would be valid to associate all the other results obtained with ROS levels? Explain this since ROS are associated to many cellular processes.
MINOR COMMENTS.
Title: Correct “indued”.
In authors ´section “Correspondence” appears twice.
Abstract:
Improve writing for “Betulinic acid (BA), a pentacyclic lupane-type triterpene, has been reported antitumor, anti-inflammatory, antiangiogenic, and anti-fibrotic effects”.
Correct grammar for “The present study aims to investigate the effect of BA on DOX-induced cardiac hypertrophy and apoptosis in vitro, H9c2 cells. were treated with BA (0.1 -1 μM) and DOX (1 μM) for 24 h.”
Improve the following phrase: “The H9c2 cells were induced by DOX (1 μM)”
The following phrase is confusing and it is necessary improve the grammar.
“The present study aims to investigate the effect of BA on DOX-induced cardiac hypertrophy and apoptosis in vitro, H9c2 cells. were treated with BA (0.1 -1 μM) and DOX (1 μM) for 24 h. The H9c2 cells were induced by DOX (1 μM) in the presence or absence of BA (0.1 -1 μM) and incubated for 24 h.”
Improve the writing for “Furthermore, BA inhibited DOX-induced the production of intracellular reactive oxygen species (ROS).”
Justify the reason for mention of ROS in results section since conditions associated to them have not been suggested.
Improve writing for “Therefore, pretreatment with BA may be potential therapies targeting cardiac hypertrophy and apoptosis leading to sudden heart failure”.
INTRODUCTION SECTION.
Improve writing for “However, administration of DOX limited by side effects like”.
The following phase is repetitive “Cardiac pathological hypertrophy is characterized by an increase in cell size including physiological and pathological hypertrophy”.
Similar observation is for the following sentence “Activation of the cardiac hypertrophy markers such as atrial natriuretic peptide (ANP), brain natriuretic peptide (BNP), β-myosin heavy chain (β-MHC), and α-skeletal actin by increased by cardiac hypertrophy”.
Section 2.1. It is indicated that cells were incubated with DOX 1 uM after pretreatment with BA but it is indicated also incubation with DOX 0.5 uM. Please clarify.
At the end of this section is stated that BA treatment regulates expression of cardiac hypertrophy marker genes but results are concern to cell viability.
Correct writing for: “Figure.1.”; “Fig.2A.”, verify for all figure legends and citations for all other figures.
Insert the adequate reference in line 143.
RESULTS SECTION.
The graphs have a poor resolution.
MATERIALS AND METHODS.
Statistical analysis. In this section text size is different.
Author Response
I am very much thankful to the reviewers for their deep and thorough review. I hope my revision has improved the paper to a level of their satisfaction. I revised the text in the Abstract, Introduction, Materials and Methods, Results, Discussion, and reference (red colored part).
Response to Reviewer Comments
The purpose of this work was investigate the effect of betulinic acid (BA) on doxorubicin (DOX)-induced cardiac hypertrophy and apoptosis in H9c2 (rat myoblastic) cells through analyses of cell viability, cell area surface, expression (western blot and mRNA) of hypertrophy markers, MAPK signaling (p38, ERK, and JNK expression), reactive oxygen species, transcription factor GATA4, calcineurin pathway and apoptosis process.
Data obtained support the following conclusion: BA protects H9c2 cells from DOX-induced hypertrophy and apoptosis at least partly through the inhibition of ROS/MAPK and GATA-4/calcineurin/NFAT signaling pathway.
According to these results BA may be considered as a new potential agent for alleviate cardiac hypertrophy and ventricular dysfunction that can lead to heart failure.
Major Comment.
OBSERVATIONS.
Comment 1: Why the authors study calcineurin/NFAT pathway? Cardiac hypertrophy is regulated by different signaling pathways as calcineurin/NFAT pathway and insulin-like growth factor-I–phosphatidylinositol 3-kinase–Akt/protein kinase and in this process participates also transcription factors such as NFAT, myocyte enhancer factor-2 (MEF-2), GATA4/6 and NFκB.
Response 1:
As suggested by the reviewer, cardiac hypertrophy is regulated by various signaling pathways such as the calcineurin/NFAT pathway and insulin-like growth factor I–phosphatidyl inositol 3-kinase–Akt/protein kinase. The reason is as follows: In many previous studies, H9c2 cardiomyocytes are known to induce cell hypertrophy through the calcineurin/NFAT-3 signaling pathway. Path ‘calcineurin/NFAT’ pathway was essentially identified in most studies(reference1,2).
Therefore, in this paper, we only confirmed the effects of calcineurin/NFAT pathway, GATA4/6, and NFκB, which are representative H9c2 cell hypertrophy mechanisms.
However, as suggested by the reviewer, we believe that further research on various paths will be needed, including the insulin-like grouth factor-I–phospatidylinositol 3-kinase–Akt/protein kinase pathways, and further studies will be conducted to supplement the limited parts of this paper.
- Kevyn E Merten et al., Zinc inhibits doxorubicin-activated calcineurin signal transduction pathway in H9c2 embryonic rat cardiac cells. Exp Biol Med (Maywood). 2007. 232(5):682-9.
- Chung-Jung Liu et al., Lipopolysaccharide induces cellular hypertrophy through calcineurin/NFAT-3 signaling pathway in H9c2 myocardiac cells. Chung-Jung Liu Mol Cell Biochem. 2008, 313(1-2):167-78.
Comment 2: Quantifications of the following figures is needed: figure 1C (cell live images), figure 2A (cell surface images), figures 2B and 2C (western blot images), figure 5A (western blot of GATA4 forms); Figures 6A, B and C (western blot of calcineurin, NFAT3 and HDAC1/ KLF4, respectively), and figures 7A and B (western blot of apoptosis markers).
Response 2: As suggested, we have added graphs for all pictures.
Comment 3: Western blots images for ANP y BNP show that BA 1 uM/Dox 1 uM treatments reduce their expression levels even below of control cells. Explain these facts and their possible consequences.
Response 3:
The expression level of ANP was similar in BA 1 uM treatment group and control group. However, in the case of the expression level of BNP, it can be seen that the BA 1 uM treatment group showed a smaller amount than the control group. This is thought to be because BA (alone treatment) inhibits BNP production. Therefore, further research is needed on the effect of BA alone treatment on BNP production in the future. Also, such cases can be found in other papers (1,2).
- Chin-Hu Lai et al., β-catenin/LEF1/IGF-IIR Signaling Axis Galvanizes the Angiotensin-II- induced Cardiac Hypertrophy. Int. J. Mol. Sci. 2019, 20(17), 4288.
- Yu-Lan Yeh et al., LPS‐enhanced IGF‐IIR pathway to induce H9c2 cardiomyoblast cell hypertrophy was attenuated by Carthamus tinctorius extract via IGF‐IR activation. Environmental Toxicology. 2020;35:145–151.
Comment 4: Section 2.3. Respect to ROS and MAPK signaling pathway the authors indicate the following: “Thus, these results suggesting that BA inhibits cardiac hypertrophy by blocking ROS-dependent MAPK signaling.” Why MAPK signaling changes were associated with changes in ROS levels? It would be valid to associate all the other results obtained with ROS levels? Explain this since ROS are associated to many cellular processes.
Response 4: ROS can activate the MAPK signaling pathway, and this process is known to play a pivotal role in cardiovascular disease. In addition, the MAPK signaling cascade is a downstream event of ROS production, which is deeply involved in cell proliferation and cell death. Although not confirmed in this paper, previous studies have confirmed that ROS and MAPK are closely related to each other in damage to H9c2 cells by doxorubicin, and this pathway is very important. In the future, we are planning a study to confirm this in depth and clarity.
- RUNMIN GUO et al., Hydrogen sulfide attenuates doxorubicin-induced cardiotoxicity by inhibition of the p38 MAPK pathway in H9c2 cells. INTERNATIONAL JOURNAL OF MOLECULAR MEDICINE 31: 644-650, 2013.
- Chun-Yan Jian et al., Naringin protects myocardial cells from doxorubicin-induced apoptosis partially by inhibiting the p38MAPK pathway. Mol Med Rep. 16(6): 9457–9463, 2017.
- Ivana Sirangelo et al., Vanillin Prevents Doxorubicin-Induced Apoptosis and Oxidative Stress in Rat H9c2 Cardiomyocytes. Nutrients. 12(8): 2317, 2020.
MINOR COMMENTS.
Comment 1: Title: Correct “indued”.
Response 1: As suggested, we corrected it in title. “Betulinic acid protects DOX-triggered cardiomyocyte hypertrophy response through the GATA-4 /calcineurin/NFAT pathway”
Comment 2: In authors ´section “Correspondence” appears twice.
Response 2: As suggested, we removed one.
Abstract.
Comment 3: Improve writing for “Betulinic acid (BA), a pentacyclic lupane-type triterpene, has been reported antitumor, anti-inflammatory, antiangiogenic, and anti-fibrotic effects”.
Response 3: As suggested, we corrected that (red color).
“Betulinic acid (BA), a pentacyclic lupane-type triterpene, has been reported antitumor, anti-inflammatory, antiangiogenic, and anti-fibrotic effects” → “Betulinic acid (BA) is a pentacyclic-cyclic lupane-type triterpene that has been reported to have anti-bacterial, anti-inflammatory, anti-vascular neogenesis and anti-fibrotic effects”
Comment 4: Correct grammar for “The present study aims to investigate the effect of BA on DOX-induced cardiac hypertrophy and apoptosis in vitro, H9c2 cells. were treated with BA (0.1 -1 μM) and DOX (1 μM) for 24 h.”
Response 4: As suggested, we corrected that (red color).
“The present study aims to investigate the effect of BA on DOX-induced cardiac hypertrophy and apoptosis in vitro, H9c2 cells. were treated with BA (0.1 -1 μM) and DOX (1 μM) for 24 h.” → “The present study aims to investigate the effect of BA on DOX-induced cardiomyocyte hypertrophy and apoptosis in vitro, H9c2 cells”
Comment 5: Improve the following phrase: “The H9c2 cells were induced by DOX (1 μM)”
Response 5: As suggested, we corrected that (red color).
“The H9c2 cells were induced by DOX (1 μM)” → “The H9c2 cells were stimulated with DOX (1 µM)”
Comment 6: The following phrase is confusing and it is necessary improve the grammar.
“The present study aims to investigate the effect of BA on DOX-induced cardiac hypertrophy and apoptosis in vitro, H9c2 cells. were treated with BA (0.1 -1 μM) and DOX (1 μM) for 24 h. The H9c2 cells were induced by DOX (1 μM) in the presence or absence of BA (0.1 -1 μM) and incubated for 24 h.”
Response 6: As suggested, we corrected that (red color).
“The present study aims to investigate the effect of BA on DOX-induced cardiac hypertrophy and apoptosis in vitro, H9c2 cells. were treated with BA (0.1 -1 μM) and DOX (1 μM) for 24 h. The H9c2 cells were induced by DOX (1 μM) in the presence or absence of BA (0.1 -1 μM) and incubated for 24 h.”
→ “The present study aims to investigate the effect of BA on DOX-induced cardiomyocyte hypertrophy and apoptosis in vitro, H9c2 cells. The H9c2 cells were stimulated with DOX (1 µM) in the presence or absence of BA (0.1 -1 μM) and incubated for 24 h.”
Comment 7: Improve the writing for “Furthermore, BA inhibited DOX-induced the production of intracellular reactive oxygen species (ROS).”
Justify the reason for mention of ROS in results section since conditions associated to them have not been suggested.
Response 7: I don't know if we understand what the reviewer suggested. We think you mentioned the conditions for ROS measurement in the part of Abstract, so I modified it accordingly.
“Furthermore, BA inhibited DOX-induced the production of intracellular reactive oxygen species (ROS).”
→ “As a result of measuring the change in ROS generation using DCF-DA, BA significantly inhibited DOX-induced the production of intracellular reactive oxygen species (ROS), when BA was treated at over concentration of 0.1 µM.”
Comment 8: Improve writing for “Therefore, pretreatment with BA may be potential therapies targeting cardiac hypertrophy and apoptosis leading to sudden heart failure”.
Response 8: As suggested, we corrected that (red color).
“Therefore, pretreatment with BA may be potential therapies targeting cardiac hypertrophy and apoptosis leading to sudden heart failure” → “Therefore, BA can be a potential treatment for cardiomyocyte hypertrophy and apoptosis that lead to sudden heart failure.”
INTRODUCTION SECTION.
Comment 9: Improve writing for “However, administration of DOX limited by side effects like”.
Response 9: As suggested, we corrected that (red color).
“However, administration of DOX limited by side effects like” → “However, administration of doxorubicin can be limited by side effects like”
Comment 10: The following phase is repetitive “Cardiac pathological hypertrophy is characterized by an increase in cell size including physiological and pathological hypertrophy”.
Response 10: As suggested, we corrected that (red color).
“Cardiac pathological hypertrophy is characterized by an increase in cell size including physiological and pathological hypertrophy” → “Cardiac hypertrophy is characterized by an increment in cardiomyocyte size including physiological and pathological hypertrophy.
Comment 11: Similar observation is for the following sentence “Activation of the cardiac hypertrophy markers such as atrial natriuretic peptide (ANP), brain natriuretic peptide (BNP), β-myosin heavy chain (β-MHC), and α-skeletal actin by increased by cardiac hypertrophy”.
Response 11: As suggested, we corrected that (red color).
“Activation of the cardiac hypertrophy markers such as atrial natriuretic peptide (ANP), brain natriuretic peptide (BNP), β-myosin heavy chain (β-MHC), and α-skeletal actin by increased by cardiac hypertrophy” → “Atrial natriuretic peptide (ANP), brain natriuretic peptide (BNP), β-myosin heavy chain (β-MHC), and α-skeletal actin are well known cardiac hypertrophy biomarkers.”
Comment 12: Section 2.1. It is indicated that cells were incubated with DOX 1 uM after pretreatment with BA but it is indicated also incubation with DOX 0.5 uM. Please clarify.
Response 12: Thank you, and we have corrected it. The content of section 2.1 is misrepresented. Corrected again exactly.
Comment 13: At the end of this section is stated that BA treatment regulates expression of cardiac hypertrophy marker genes but results are concern to cell viability.
Response 13: As suggested, we corrected that (red color).
→ “Thus, BA treatment mitigates the cytotoxicity of H9c2 under DOX conditions.”
Comment 14: Correct writing for: “Figure.1.”; “Fig.2A.”, verify for all figure legends and citations for all other figures.
Response 14: Thank you, and we have corrected it (red color).
Comment 15: Insert the adequate reference in line 143.
Response 15: Thank you, and we added a reference [32].
RESULTS SECTION.
Comment 16: The graphs have a poor resolution.
Response 16: Thank you, As suggested, we improved the graphs resolution.
MATERIALS AND METHODS.
Comment 17: Statistical analysis. In this section text size is different.
Response 17: Thank you, As suggested, we have changed the text size for the "Statistical analysis" section to be the same.
Round 2
Reviewer 1 Report
My concerns have been addressed.